# Multi-Walled Carbon Nanotubes for Magnetic Solid-Phase Extraction of Six Heterocyclic Pesticides in Environmental Water Samples Followed by HPLC-DAD Determination

**DOI:** 10.3390/ma13245729

**Published:** 2020-12-15

**Authors:** Jiping Ma, Liwei Hou, Gege Wu, Liyan Wang, Xiaoyan Wang, Lingxin Chen

**Affiliations:** 1School of Environmental & Municipal Engineering, Qingdao University of Technology, Qingdao 266033, China; hlw1120@163.com (L.H.); 18354215495@163.com (G.W.); 2Key Laboratory of Coastal Environmental Processes and Ecological Remediation, Research Centre for Coastal Environmental Engineering and Technology, Yantai Institute of Coastal Zone Research, Chinese Academy of Sciences, Yantai 264003, China; lywang@yic.ac.cn (L.W.); wangxy@yic.ac.cn (X.W.); 3School of Pharmacy, Binzhou Medical University, Yantai 264003, China

**Keywords:** mag-multi-walled carbon nanotubes (mag-MWCNTs), magnetic solid-phase extraction (MSPE), heterocyclic pesticides, water samples

## Abstract

Magnetic multi-walled carbon nanotubes were prepared as magnetic solid-phase extraction (MSPE) adsorbent for the enrichment of six heterocyclic pesticides in environmental water samples, including imidacloprid, triadimefon, fipronil, flusilazole, chlorfenapyr and fenpyroximate. Then six heterocyclic pesticides were separated and determined by high-performance liquid chromatography-diode-array detector (HPLC-DAD). Major factors influencing MSPE efficiency, including the dose of mag-multi-walled carbon nanotubes (mag-MWCNTs), extraction time, solution pH, salt concentration, type and volume of eluent and desorption time were investigated. Under the optimized conditions, the enrichment factor of the method reached to 250. The linearity was achieved within 0.05–10.0 μg/L for imidacloprid and chlorfenapyr, 0.10–10.0 μg/L for fipronil, flusilazole, triadimefon and fenpyroximate. Limits of detection were in the range of 0.01–0.03 μg/L. Good precision at three spiked levels were 1.1–11.2% (intra-day) and 1.7–11.0% (inter-day) with relative standard deviation of peak area, respectively. The developed method was utilized to analyze tap water, river water and reservoir water samples and recoveries at three spiked concentration levels ranged from 72.2% to 107.5%. The method was proved to be a convenient, rapid and practical method for sensitive determination of heterocyclic pesticides.

## 1. Introduction

Heterocyclic pesticides have become one of the fast-growing pesticides in recent years for their higher efficiency, lower toxicity and lower residue than traditional pesticides [1]. Imidacloprid, triadimefon, fipronil, flusilazole, chlorfenapyr and fenpyroximate take great market shares and have been widely applied in rice, vegetable, fruit trees and ornamental plants to control weeds and pests. However, they still exhibit toxicity and other undesirable side effect on non-target organisms [1,2]. For example, fipronil has nerve and reproductive development toxicity to human body and has been prohibited from use in some countries. Chlorfenapyr is toxic to aquatic organisms and may cause long-term adverse effects in the aquatic environment. Therefore, inappropriate use of heterocyclic pesticides could cause environmental pollution and do harm to human health. It is imperative to develop fast, sensitive, accurate and simple methods to analyze the trace but widely distributed heterocyclic pesticides in complex environments, to obtain information such as the presence, migration and transformation about them, and finally to accurately assess the effects that they may bring to the environment and human beings.

Several analytical methods have been applied to the determination of heterocyclic pesticides, including high performance liquid chromatography-ultraviolet or diode-array detector (HPLC-UV/DAD) [3,4,5], high performance liquid chromatography-tandem mass spectrometry (HPLC-MS/MS) [6], gas chromatography [7], gas chromatograph-mass spectrometry (GC-MS) [8,9,10] and capillary electrophoresis [11]. However, low concentration levels of targets and high matrices level hinder the direct determination of heterocyclic pesticides in environmental samples. Therefore, preconcentration techniques about enrichment of ppb level heterocyclic pesticides from water samples are necessary. So far, many kinds of pretreatment techniques have been reported, such as, solid-phase extraction (SPE) [11,12,13,14], solid phase microextraction (SPME) [9,15,16], dispersive liquid–liquid microextraction (DLLME) [5], and cloud point extraction (CPE) [17]. Among them, SPE is regarded as a powerful tool to enrich organic compounds with advantages of high recovery, high enrichment factor and less consumption of organic solvents. However, traditional SPE process is tedious and SPE cartridges are easy to be blocked by suspended particles in water samples.

Recently, magnetic solid-phase extraction (MSPE) [18] has been well developed as a novel SPE method. In MSPE mode, small amounts of magnetic adsorbents are dispersed in a sample solution to adsorb target analytes, then they are rapidly collected under magnetic fields. Compared with traditional SPE, MSPE is rapid and magnetic adsorbents could be easily separated from water solution without complicated device. Since adsorption materials play a very important role in MSPE, current researches on MSPE mainly focused on the development of new adsorbents. Up to now, some magnetic particles have been reported as MSPE adsorbents [19], such as magnetic C_18_ [20], magnetic ionic liquid [21,22], magnetic multi-walled carbon nanotubes [23], magnetic MOFs [24] and magnetic molecularly imprinted polymers [25]. Peng et al. [22] used ionic liquid coated microspheres Fe_3_O_4_@[C_8_mim][PF_6_] for MSPE of clofentezine and chlorfenapyr in environmental water samples. Our group [24] prepared magnetic metal-organic frameworks (MOFs) Fe_3_O_4_@SiO_2_/MIL-101 as MSPE adsorbent for the enrichment and determination of four kinds of pyrazole/pyrrole pesticides in environmental water samples.

Multi-walled CNTs (MWCNTs) were assembled by several layers of rolled graphite sheets [26]. They have demonstrated great application potentials in sensor [27], catalysis [28] water treatment [29]. Their high surface areas lead to excellent adsorption capabilities, so they have been used as SPE adsorbents to extract various analytes. In recent years, we have employed MWCNTs as adsorbents for SPE of 16 polycyclic aromatic hydrocarbon (PAHs) [30] and pyrazole/pyrrole pesticides [14] in environmental water samples. The extraction efficiencies of PAHs and pyrazole/pyrrole pesticides with MWCNTs were satisfactory. However, the extraction time was long. Magnetic multiwall carbon nanotube composites (mag-MWCNTs) are hybrids of magnetite Fe_3_O_4_ and MWCNTs, combining advantages of MWCNTs and magnetic particles. They have been employed as adsorbents for determination of estrogens [31], phthalate acid esters [32,33], aromatic amines [34], herbicides [23], metal ions [35] and so on.

In this work, magnetic MWCNTs were prepared by solvothermal method and characterized by Fourier transform infrared (FT-IR) and transmission electron microscope (TEM). Then the prepared mag-MWCNTs were utilized as MSPE adsorbent for enrichment of six widely used heterocyclic pesticides (imidacloprid, triadimefon, fipronil, flusilazole, chlorfenapyr and fenpyroximate, as shown in Figure 1). The simultaneous determination method of six heterocyclic pesticides in environmental water samples was developed by MSPE coupled with HPLC-DAD. Major factors that possibly influence MSPE efficiency were investigated, including the dose of mag-MWCNTs, extraction time, solution pH, salt concentration, type and volume of eluent and desorption time. The MSPE-HPLC-DAD method was validated. The linearity, precision and accuracy of the method were obtained. The method was employed for the simultaneous determination of six heterocyclic pesticides in tap, river and reservoir water samples.

## 2. Experimental

### 2.1. Chemical Reagents and Solutions

Carboxylic MWCNTs (>50 nm inner diameter, 10–20 μm length) were purchased from Chinese Academy of Sciences, Chengdu Organic Chemistry Co., Ltd. (Chengdu, China). Ferric chloride hexahydrate was purchased from Sinopharm Chemical Reagent Co. Ltd. (Shanghai, China). Ethylene glycol was purchased from Fuyu Fine Chemical Co., Ltd. (Tianjin, China). Sodium acetate and anhydrous ethanol were purchased from Aibi Chemical Reagent Co., Ltd. (Shanghai, China). Anhydrous sodium chloride was purchased from Jiangsu Powerful Function Chemical Co., Ltd. (Suzhou, China). All chemicals were at least of analytical grade (purity is higher than 99.7%). Solvents are of HPLC grade including methanol, acetonitrile and acetone, which were obtained from TEDIA (Fairfield, OH, USA). HPLC grade of ethyl acetate was purchased from Honeywell (Morristown, NJ, USA). Ultrapure water (18.2 MΩ cm) was produced by a model Synergy 185 ultra-pure water system (Millipore, Boston, MA, USA).

Four analytical standards of fipronil (>98.3% purity), flusilazole (>99% purity), chlorfenapyr (>98.6% purity) and fenpyroximate (>98.4% purity) were purchased from the Testing Center of the Shanghai Pesticide Research Institute (Shanghai, China). A stock solution of each analyte at 1000 mg/L was prepared by dissolving solid pesticides standards in methanol. The other two analytical standard solution (100 mg/L, dissolved in methanol) of imidacloprid and triadimefon were also purchased from the Testing Center of the Shanghai Pesticide Research Institute (Shanghai, China). These stock solutions were stored at −20 °C in the dark.

The surface water samples were collected from Qingdao Dagu river and Qingdao Jihongtan Reservoir. The tap water samples were collected from our laboratory. All water samples were filtered through 0.45 μm membranes and stored in darkness at 4 °C before analysis.

### 2.2. Apparatus

In this experiment, Agilent 1100 series high performance liquid chromatography with a DAD detector was used to determine the concentrations of six heterocyclic pesticides. Agilent ZORBAX SB-C_18_ (4.6 × 150 mm, 5 μm) (Santa Clara, CA, USA) was used for the separation of heterocyclic pesticides at room temperature. The sample injection volume was 10 µL. DAD absorbance was monitored at 270 nm (for imidacloprid) and 215 nm (for other five compounds). The mobile phase was a mixture of acetonitrile (A) and water (B). The gradient elution conditions were as follows: 23% A (0 min, hold 1.5 min), 50% A (12 min, hold 8 min), 75% A (30 min, hold 5 min). The flow rate was 1 mL/min. The analyzed six heterocyclic pesticides can be well separated from each other.

FEI Tecnai G2 F20 transmission electron microscope (TEM, FEI, Hillsboro, OR, USA) and Fourier transform infrared (FT-IR, PerkinElmer Frontier, Waltham, MA, USA) were used to characterize the mag-MWCNTs.

### 2.3. Preparation of Mag-MWCNTs

Magnetic MWCNTs were synthesized by a reported solvothermal method [23], as schematically shown in Figure 2.

### 2.4. MSPE Process

Before extraction, 40 mg mag-MWCNTs were subsequently activated by 5.0 mL of methanol, 5.0 mL of ethyl acetate and 10.0 mL of ultra-pure water. Then the dispersed mag-MWCNTs were gathered by a strong magnet attached outside of the beaker, and the water was decanted. After that, 100 mL spiked water sample was added into the beaker and stirred for 8 min. Then, mag-MWCNTs were collected by the external magnet and the supernatant water was decanted. Remained mag-MWCNTs were washed twice with 4 mL of ethyl acetate. All ethyl acetate eluent was combined and concentrated by the rotary evaporators at room temperature, then diluted by 0.4 mL acetonitrile, and filtered through a 0.45-μm nylon membrane for HPLC. The MSPE process is schematically shown in Figure 2.

## 3. Results and Discussion

### 3.1. Characterization of the Mag-MWCNTs

The surface characterizations of mag-MWCNTs were investigated by TEM technique. As shown in Figure 3A, the micrograph of mag-MWCNTs revealed that the spherical Fe_3_O_4_ particles were linked with the tubular MWCNTs, and Figure 3B showed that the nanoparticle size of Fe_3_O_4_ was about 100 nm. As shown in Figure 4, the sample solution became turbid due to the uniformly dispersed mag-MWCNTs (Figure 4A). When the external magnetite was attached to the outside of the vial, the mag-MWCNTs adhered together at the vial’s inner side, and the water solution became clear and transparent (Figure 4B), suggesting the fast and simple process of magnetic separation.

The FT-IR spectra of Fe_3_O_4_ (a), and MWCNTs (b) and mag-MWCNTs (c) were shown in Figure 5. The absorption peak at 580 cm^−1^ in Figure 5a might be attributed to the Fe–O–Fe vibration in Fe_3_O_4_. In Figure 5b, the absorption peak at 3462 cm^−1^ was ascribed to O–H stretching vibration, the peak at 1618 cm^−1^ and 1448 cm^−1^ were due to C=C of the benzene ring in MWCNTs. All the above characteristic peaks appeared in Figure 5c, indicating the mag-MWCNTs composites were successfully synthesized. The results were consistent with reported patterns of magnetic MWCNTs [35].

### 3.2. Optimization of MSPE Conditions

Some parameters would influence the MSPE performance, including (i) adsorption of the pesticides: amount of mag-MWCNTs, extraction time, solution pH, salt concentration; (ii) release of the pesticides by elution: type and volume of desorption solvent, desorption time. These factors were investigated to obtain the best enrichment performance by spiking ultrapure water samples at 4 µg/L.

#### 3.2.1. Effect of Amount of Mag-MWCNTs

An insufficient amount of adsorbent would cause the loss of the analytes, whereas a higher amount would increase the cost of the analytical procedure. The adsorbent amount was examined from 20 to 50 mg. As shown in Figure 6A, the extraction recoveries for six heterocyclic pesticides increased with the adsorbent amount increased from 20 to 40 mg; however, the extraction efficiencies at 50 mg were almost the same efficiencies at 40 mg. The reason might be that the separation procedure becomes difficult with the increase of adsorbent amount. Satisfactory extraction efficiencies could be obtained when the amount of the mag-MWCNTs was 40 mg.

#### 3.2.2. Effect of Extraction Time

Insufficient extraction time would influence the establishment of adsorption equilibrium, so extraction time is a key factor. As shown in Figure 6B, extraction recoveries of six heterocyclic pesticides were significantly improved to above 82%, with the extraction time increasing to 8 min. Therefore, 8 min was chosen as the experimental extraction time.

#### 3.2.3. Effect of Solution pH

The pH value has a great influence on the adsorption performance of the adsorbent. Effects of sample pH on extraction recoveries were investigated. As shown in Figure 6C, when pH ranging from 2 to 7, extraction recoveries of five heterocyclic pesticides were all above 77% except lower extraction recovery of fenpyroximate at pH = 2. It is likely because interactions between heterocyclic pesticides and mag-MWCNTs were hydrogen bonds and π-π interactions. Thus, solution pH had little effect on adsorption. At pH 6.0, extraction recoveries of all heterocyclic pesticides were higher than 80%, which could satisfy the analytical requirement. Therefore, sample solution pH was adjusted at pH 6.0 in the following procedures.

#### 3.2.4. Effect of Salt Concentration

Generally, the salting-out effect resulting from the additional salt could decrease the solubility of analytes. What is more, the diffusion rate of analytes from aqueous to adsorbents can be adjusted by the salt in water samples. To investigate effect of salt concentration on extraction of six heterocyclic pesticides, different concentrations of NaCl (0, 0.5, 1, 5 and 10%, *w*/*v*) were investigated. As shown in Figure 6D, the highest recoveries of heterocyclic pesticides were obtained when no NaCl was added. This phenomenon might be due to the addition of NaCl inhibiting the mass transfer from the aqueous to solid phases by increasing the viscosity and the density of the solution. Therefore, no NaCl was added in the following experiments.

#### 3.2.5. Effect of Type of Eluent

Elution is a critical step in the MSPE and plays a significant role in the extraction performance. Different solvents will result in different extraction efficiencies. Four solvents (methanol, acetonitrile, acetone, and ethyl acetate) were used to investigate the effect of desorption solvents on extraction recoveries. Figure 6E showed that acetonitrile and ethyl acetate exhibited higher extraction recoveries than methanol and acetone. The best extraction recovery was obtained by using ethyl acetate, which could make the recoveries of all heterocyclic pesticides higher than 80%. Therefore, ethyl acetate was chosen as the desorption solvent for further experiments.

#### 3.2.6. Effect of Volume of Eluent

The volume of eluent significantly affects the desorption efficiency and analytical sensitivity of the developed method. 5, 8, 12, and 15 mL of ethyl acetate were used to investigate the effect of the eluent volume. The results shown in Figure 6F revealed that 8 mL of ethyl acetate was sufficient to desorb the adsorbed analytes. Increasing the volume of ethyl acetate could not significantly increasing the extraction efficiencies. Therefore, 8 mL of ethyl acetate was selected for desorption in further experiments.

#### 3.2.7. Effect of Desorption Time

Desorption time was investigated at 2, 4 and 6 min. It could be found in Figure 6G that the recoveries increased obviously for imidacloprid, triadimefon, flusilazole and fenpyroximate when desorption time increasing from 2 min to 4 min, and extraction recoveries of heterocyclic pesticides were above 80%, which could satisfy the analytical requirement. Therefore, desorption time was fixed at 4 min.

The possible adsorption mechanism is dominated by π-π interaction between the benzene rings in MWCNT and the pesticides. As seen in Figure 6, mag-MWCNT exhibited lower extraction efficiencies to the pesticide with higher polarity (i.e., imidacloprid) due to the electron drawing groups decrease the electron cloud density of the benzene ring in imidacloprid, leading to the weaker π-π interaction. Thus, the MWCNT exhibited different extraction behavior towards different pesticides.

### 3.3. Validation of the MSPE-HPLC Method

Under the optimized conditions, the linearity, limits of detection (LODs), limits of quantification (LOQs) were used to investigate the sensitivity of the proposed method, the precision and reproductivity of the method were determined by intra-day and inter-day RSDs. The calibration curve of each heterocyclic pesticide was constructed by plotting the peak areas (y) of heterocyclic pesticide after preconcentration versus the concentration of the analyte (x). Satisfactory correlation coefficients (between 0.9977 and 0.9986) of the heterocyclic pesticides were obtained within the linearity range of 0.05–10 μg/L for imidacloprid and chlorfenapyr, 0.10–10.0 μg/L for fipronil, flusilazole, triadimefon and fenpyroximate (Table 1). The LODs and LOQs were calculated as the concentration of heterocyclic pesticide at signal-to-noise ratios of 3 and 10, respectively. The results showed that the LODs and LOQs (Table 1) for the heterocyclic pesticides ranged from 0.01–0.03 μg/L and 0.04–0.12 μg/L, respectively. The LODs for six heterocyclic pesticides are lower than the maximum residue level (MRL) of 1 μg/L in European Union surface water regulation, which was used for the source of drinking water (75/440/EEC) [36] and 0.1 μg/L in European Union drinking water regulation (98/83/EC) [37]. The intra-day and inter-day relative standard deviations (RSDs, %) were measured with heterocyclic pesticides spiked at 0.2 μg/L, 1 μg/L and 8 μg/L in ultrapure water. The intra-day RSDs were obtained by the spiked recoveries from six sample solutions within one day. And the inter-day RSDs were determined from six contiguous days. The results showed that the intra- and inter-day RSDs were 1.1–11.2% and 1.7–11.0%, respectively (Table 2). This demonstrated the good reproducibility of the method.

### 3.4. Method Application to Real Water Samples

The proposed method had been validated by three water samples from tap water, river water and reservoir water. HPLC chromatograms achieved for the reservoir water sample with and without spiking after MSPE were displayed in Figure 7. As seen, six heterocyclic pesticides were not detected in these samples (Figure 7a and Table 3), and significant peaks were appeared after spiking (Figure 7b). The recoveries were obtained by spiking real water samples at 0.2 μg/L, 1 μg/L and 8 μg/L individual pesticides. The recoveries and RSDs of three real water samples were calculated from the average of three replicates, as shown in Table 3. The recoveries of six heterocyclic pesticides ranged from 72.2–105.0% for tap water samples with RSDs of 1.6–11.9%, 81.2–99.4% for river water samples with RSD of 1.8–12.6%, and 79.8–107.5% for reservoir water samples with RSDs of 1.6–10.2%. All results showed that mag-MWCNTs were promising adsorbents for heterocyclic pesticides, and the proposed method was adaptable in complex water samples.

### 3.5. Method Performance Comparison

The obtained results were compared with those reported in previous works, including SPME [15], DLLME [5], SPE [13], CPE [17] and MSPE [22,24] (Table 4). Compared with those reported methods using HPLC-UV/DAD as a detector, this method presented lower LODs than most of the listed methods. In our previous study [13], the SPE-HPLC method using multi-walled carbon nanotubes as adsorbent showed the lowest LODs. However, the pretreatment time of conventional SPE was nearly 4 h while the pretreatment time of MSPE in this work was less than 30 min. Although the CPE-HPLC method [17] also had lower LODs, it needs expensive equipment such as centrifuge, while the MSPE in this work only needs magnets to separate mag-MWCNTs from samples. The method using SPE with UPLC-MS/MS exhibited the lowest LOD [38]. However, UPLC-MS/MS is expensive, leading to higher analytical cost. In short, the method developed in our current study possessed advantages of high sensitivity, cost-effectiveness and rapid simple magnetic separation.

## 4. Conclusions

The present study showed a practical application of mag-MWCNTs as an effective adsorbent in MSPE coupled with HPLC for the determination of six heterocyclic pesticides in environmental water samples. Good extraction efficiencies and high sensitivity with low LODs/LOQs of six heterocyclic pesticides were achieved. Additionally, the separation of mag-MWCNTs from water solutions could be achieved by an external magnet and did not require any complicated equipment. Therefore, the enrichment, separation, and release of six heterocyclic pesticides of the proposed method were demonstrated to be fast, sensitive, accurate and simple.

## Figures and Tables

**Figure 1 materials-13-05729-f001:**
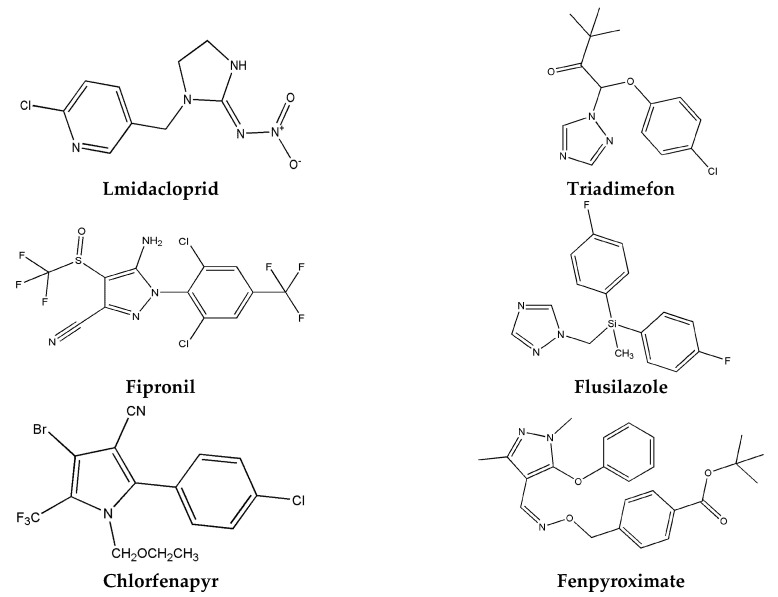
Structural formula of six heterocyclic pesticides.

**Figure 2 materials-13-05729-f002:**
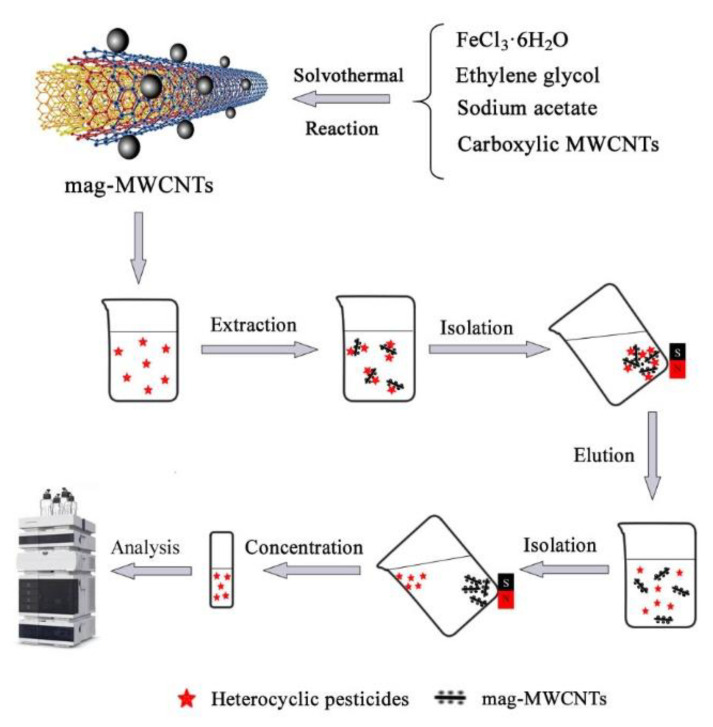
Schematic illustration of the preparation process for the mag-multi-walled carbon nanotubes (mag-MWCNTs) and the magnetic solid-phase extraction (MSPE) procedure.

**Figure 3 materials-13-05729-f003:**
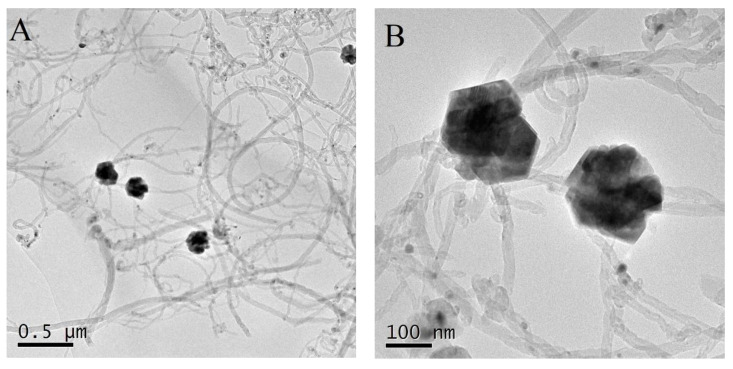
TEM image of mag-MWCNTs with different image scale: (**A**) 0.5 μm, (**B**) 100 nm.

**Figure 4 materials-13-05729-f004:**
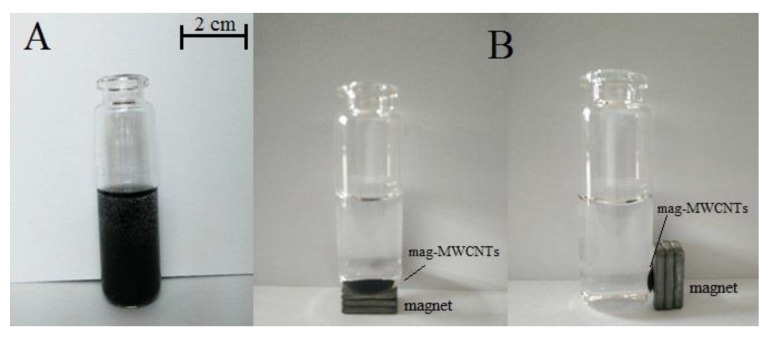
Picture of mag-MWCNTs dispersed in water (**A**) and adhered to the inner side wall of the vial (**B**).

**Figure 5 materials-13-05729-f005:**
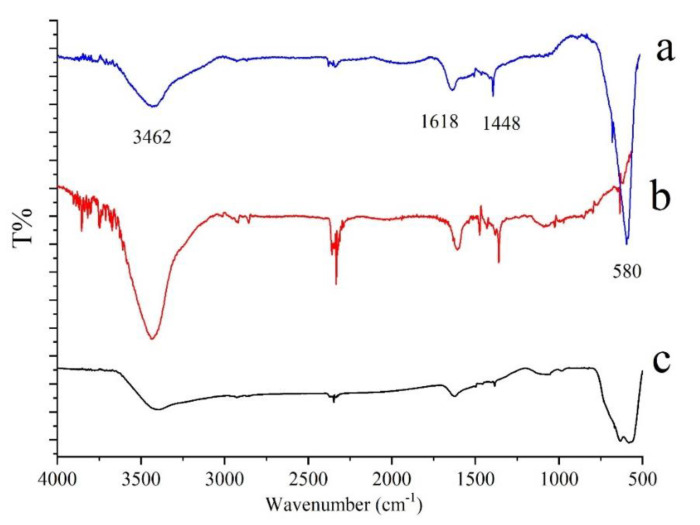
FT-IR spectra of Fe_3_O_4_ (**a**), and MWCNTs (**b**) and mag-MWCNTs (**c**).

**Figure 6 materials-13-05729-f006:**
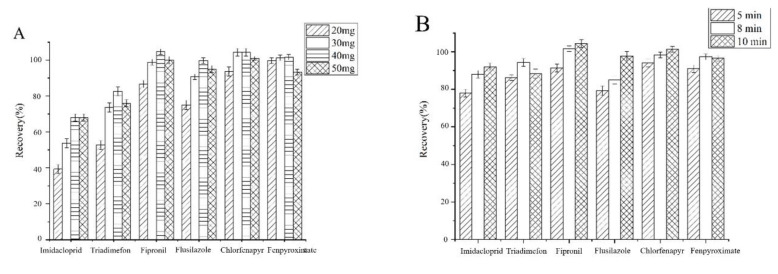
Effect of (**A**) the amount of mag-MWCNTs (**B**) extraction time, (**C**) sample pH, 6.0; (**D**) salt concentration, (**E**) type of desorption solvent, (**F**) volume of desorption solvent, and (**G**) desorption time.

**Figure 7 materials-13-05729-f007:**
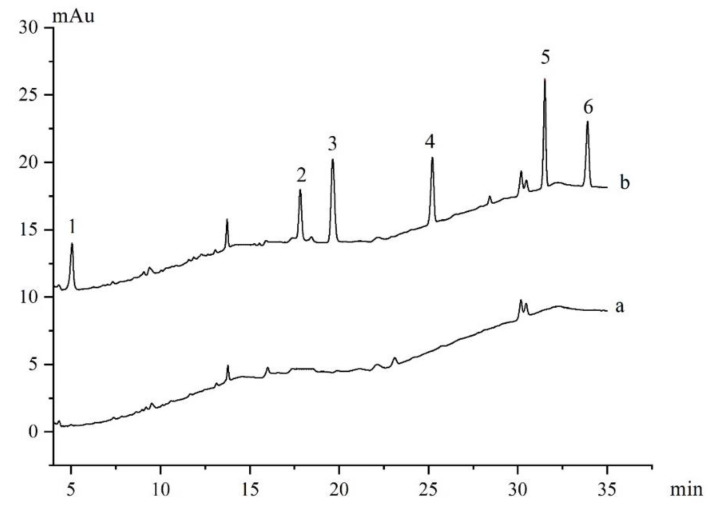
HPLC chromatograms of six heterocyclic pesticides in real reservoir water samples after MSPE without spiking (**a**) and with spiking (**b**). Peak identification: (1) Imidacloprid (2) triadimefon, (3) flusilazole, (4) fipronil, (5) chlorfenapyr, (6) fenpyroximate. Extraction conditions were the optimal condition.

**Table 1 materials-13-05729-t001:** Linear regression equations, correlation coefficients, LODs and LOQs for the six heterocyclic pesticides.

Pesticides	Regression Equation ^a^	R^2^	Linear Range (μg/L)	LODs (μg/L)	LOQs (μg/L)
Imidacloprid	y = 9.24x + 0.3	0.998	0.05–10.0	0.01	0.04
Triadimefon	y = 4.78x + 0.22	0.998	0.10–10.0	0.02	0.08
Fipronil	y = 7.98x − 0.75	0.998	0.10–10.0	0.02	0.08
Flusilazole	y = 7.45x − 0.46	0.999	0.10–10.0	0.03	0.12
Chlorfenapyr	y = 8.01x + 0.02	0.998	0.05–10.0	0.01	0.04
Fenpyroximate	y = 6.90x − 0.43	0.998	0.10–10.0	0.03	0.12

^a^ x = concentration (µg/L), y = area.

**Table 2 materials-13-05729-t002:** Intra-day and inter-day precisions (RSD, %) for MSPE-HPLC determination of the six heterocyclic pesticides.

Pesticides	Spiked (μg/L)	Intra-Day (*n* = 6)	Inter-Day (*n* = 6)
Imidacloprid	0.1	2.8	8.8
	1.0	2.2	2.0
	8.0	1.9	1.7
Triadimefon	0.1	8.8	8.4
	1.0	11.2	11.0
	8.0	2.9	2.9
Fipronil	0.1	7.3	7.7
	1.0	8.6	9.7
	8.0	1.8	2.2
Flusilazole	0.1	7.1	9.0
	1.0	4.8	8.3
	8.0	1.1	1.8
Chlorfenapyr	0.1	2.9	10.5
	1.0	3.8	7.2
	8.0	5.6	5.1
Fenpyroximate	0.1	2.0	8.2
	1.0	2.8	3.4
	8.0	2.3	2.9

**Table 3 materials-13-05729-t003:** Determination of six heterocyclic pesticides and method recoveries in real water samples.

Pesticides	Spiked (µg/L)	Tap Water		River Water		Reservoir Water	
		Found (µg/L)	Recovery (%, *n* = 3)	Found (µg/L)	Recovery (%, *n* = 3)	Found (µg/L)	Recovery (%, *n* = 3)
Imidacloprid	0	UD ^a^		UD		UD	
	0.2	0.14	72.2 ± 11.9 ^b^	0.16	82.6 ± 12.6	0.17	84.7 ± 4.4
	1.0	0.88	86.8 ± 4.7	0.81	81.2 ± 8.8	0.82	81.7 ± 10.2
	8.0	6.8	85.2 ± 5.4	6.61	82.7 ± 4.7	7.41	92.7 ± 1.6
Triadimefon	0	UD		UD			
	0.2	0.16	80.1 ± 7.1	0.19	93.5 ± 9.1	0.17	83.6 ± 5.0
	1.0	1.05	105.0 ± 2.2	0.94	94.0 ± 3.7	0.98	98.1 ± 8.3
	8.0	7.89	98.7 ± 4.0	6.95	86.8 ± 2.6	7.75	96.9 ± 3.5
Fipronil	0	UD		UD		UD	
	0.2	0.18	93.4 ± 1.6	0.16	83.5 ± 5.0	0.16	82.0 ± 9.6
	1.0	0.95	95.7 ± 2.8	0.83	83.7 ± 3.6	0.86	85.6 ± 4.0
	8.0	8.19	102.3 ± 2.5	7.95	99.4 ± 3.9	8.60	107.5 ± 3.8
Flusilazole	0	UD		UD		UD	
	0.2	0.16	79.7 ± 1.83	0.16	81.7 ± 1.8	0.163	81.7 ± 3.5
	1.0	0.88	88.6 ± 5.5	0.81	81.1 ± 3.7	0.96	95.9 ± 4.1
	8.0	7.84	98.0 ± 3.2	7.45	93.1 ± 2.2	7.44	93.0 ± 4.1
Chlorfenapyr	0	UD		UD		UD	
	0.2	0.17	85.1 ± 2.1	0.17	89.2 ± 9.6	0.16	79.8 ± 4.5
	1.0	1.00	100.4 ± 3.7	0.90	90.1 ± 9.2	0.91	90.9 ± 2.4
	8.0	8.27	103.4 ± 2.4	7.94	99.2 ± 4.9	8.3	104.4 ± 2.6
Fenpyroximate	0	UD		UD		UD	
	0.2	0.19	96.9 ± 2.3	0.17	98.0 ± 3.5	0.18	92.3 ± 8.0
	1.0	0.88	88.4 ± 1.9	0.84	84.1 ± 6.0	0.92	92.5 ± 2.9
	8.0	8.08	101.0 ± 2.8	7.75	96.8 ± 2.4	7.58	94.7 ± 7.6

^a^ Under the limits of detection. ^b^ RSD.

**Table 4 materials-13-05729-t004:** Method comparisons for analysis of heterocyclic pesticides in water samples.

Matrix	Methods	Compounds ^a^	LODs (μg/L)	Pretreatment Time	Reference
Tap water, underground water	SPME-GC-MS	1	0.08	35 min	[15]
Reservoir water, sea water	SPE-HPLC-DAD	4	0.008–0.019	≈4 h	[13]
Tap water, lake water, spring water	DLLME-HPLC-DAD	4	0.53–1.28	≈15 min	[5]
Tap water, river water	CPE-HPLC-UV	4	0.0068–0.0345	>15 min	[17]
River water	MSPE-HPLC-VW	2	0.4–0.5	>25 min	[22]
Reservoir water, sea water, river water	MSPE-HPLC-DAD	4	0.3–1.5	>25 min	[24]
Seawater and river water	SPE-UPLC-MS/MS	11	0.02–0.1 ng/L	>40 min	[38]
Tap water, river water, reservoir water	MSPE-HPLC-DAD	6	0.01–0.03	≈25 min	This work

^a^ the kinds of detected heterocyclic pesticides.

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
