# Peer review of "Multi-Walled Carbon Nanotubes for Magnetic Solid-Phase Extraction of Six Heterocyclic Pesticides in Environmental Water Samples Followed by HPLC-DAD Determination"

_materials, 2020, doi:10.3390/ma13245729_

Round 1

Reviewer 1 Report

In this paper, the authors describe the use of magnetic multi-wall CNTs to extract heterocyclic pesticides from tap water or reservoir water with shorter pretreatment time and limits of detections better, if not comparable to previously reported works. The paper is well-written and I have a few comments that can improve the paper further:

  1. The first sentences of the introduction needs references about the recent impact of heterocyclic pesticides.
  2. On page 2, line 46, "distribute" is best replaced by "distribution of".
  3. On page 2, line 55, what are values for trace/ultra-trace? For example the authors could talk either in ppm or ppb.
  4. On page 2, line 66, "time saving" is best replaced by "fast" or "rapid".
  5. On page 3, line 97, please define the acronym "I.D." as appropriate such as inner diameter.
  6. Analytical grade means what percentage purity approximately? Please put a value, usually this is >95%. The terminology is different now in different countries. So it might be better to put the approximate purity.
  7. Figure 2 contains many mistakes in spelling. Please correct. In addition, the caption is not descriptive, perhaps add the step sequence in the caption too.
  8. On page 5, line 149, no need to redefine SEM since it was already defined in the introduction section.
  9. Figure 4 needs an approximate scale if available. Please also label the magnet, and magnetic particle locations in the image. A and B are not visible. Please correct.
  10. Figure 5 is not XRD. This is FT-IR spectra. Please correct. In addition, please revise the introduction section as it also says that the authors did XRD. Also add the description of (a), (b) and (c) in the caption.
  11. The explanation on lines 193-194 on page 7 about the pH having limited effect on the adsorption needs a reference to support this. The data shown in Figure 6C is skewed by the Fenpyroximate result at a pH of 2. Since there is monotonic increase in recovery  at higher pH, why not use pH of 7 rather than 6 since 7 is the neutral pH?
  12. In sections 3.2.5 to 3.2.7, the authors use the parameters that give >80% recovery for all the pesticides. From Figure 6, it is not very clear, perhaps the authors could use a dashed line at 80% for Figure 6E-6G to mark the 80% mark as the threshold they are aiming for.
  13. Table 2 Column 1 needs to have consistent decimal places.

Reviewer 2 Report

The manuscript describes the development of an analytical method using magnetic multiwalled nanotubes as sorbents for the extraction of pesticides from water samples. The manuscript does not fit at all within the scope of materials. It is pure analytical method development for pesticides with an application to water samples. The analytical part is not new as the authors used the same approach on other analytes already. The synthesis of the MWCNTs is already reported in anther paper. There is no novelty in the manuscript and no materials content. I can only support rejection, especially for Materials.

Additional comments:

The paper has no materials content. Also see Literature citations, all analytical/chromatography papers some environmental ones not material ones

It looks like largely the same than the authors published already just exchanging analytes.

There are gross errors in the paper. The authors talk about XRD when they did FTIR… there is no XRD data and the figure is clearly FTIR. The quality of the FTIR is very poor!

The quality of the HPLC chromatogram is very poor, very odd baseline

The benchmarking against other methods is selective and based… there are many other, more preforming methods using LC/MS for these analytes.

The authors use excessive digits in numbers… and figures are lacking error bars.

Table 3: why is one pesticide in bold… also why analyzing samples that have no pesticide. Plus you should state <x not ND so state your detection limit.

Reviewer 3 Report

The authors present a work describing the use of magnetic CNT nanocomposites in analytical applications. The novelty is limited although the good analytical performances that they claim make the work more interesting. While the analytical part is detailed, the characterization of the material is not so detailed. The authors wrote that they used a published procedure but at least they should evaluate the content of the metal oxide in the nanocomposite. In the introduction (line 85) they reported an x ray analysis but I could not find it in the work. Imidacloprid is always written wrong in every scheme and figure.

Line 132: Figure 2 does not explain the solvothermal process

The effect of the salt (line 200) is negligible, I would not accept however the explanation offered.

The text is cattered with typos and difficult to read sentences that might be improved

Round 2

Reviewer 2 Report

The authors only answered the questions politely but hardly made any significant changes to the manuscript. This is still not a materials paper, no effort is made ot frame it as such (intro or references). It is still of low quality (e.g. FT-IR spectra still look poor, figures do not have error bars etc). The comparisons are still biased (LC/MS in most of the Asia, EU, USA this is the standard for trace contaminants and needs to at least be mentioned! we are in 2020! you can write that it is expensive etc.. but you cannot just ignore what is the state of the art!). English did not improve instead they added new issues by not even correctly making the changes from the other reviewers (see "repaid" instead of "rapid"). There is a clear lack of attention to detail!!!!

Reviewer 3 Report

The authors have mostly fulfilled the requests. The manuscript can be accepted

Author Response

Thanks for the reviewer's comment.